# Molecular Mechanisms of lncRNAs in the Dependent Regulation of Cancer and Their Potential Therapeutic Use

**DOI:** 10.3390/ijms23020764

**Published:** 2022-01-11

**Authors:** Carlos García-Padilla, Ángel Dueñas, Virginio García-López, Amelia Aránega, Diego Franco, Virginio Garcia-Martínez, Carmen López-Sánchez

**Affiliations:** 1Department of Experimental Biology, University of Jaen, 23071 Jaen, Spain; aduenle@gmail.com (Á.D.); Aaranega@ujaen.es (A.A.); dfranco@ujaen.es (D.F.); 2Department of Human Anatomy and Embryology, University of Extremadura, 06006 Badajoz, Spain; garcialopez@unex.es (V.G.-L.); virginio@unex.es (V.G.-M.); 3Institute of Molecular Pathology Biomarkers, University of Extremadura, 06006 Badajoz, Spain; 4Fundación Medina, 18016 Granada, Spain

**Keywords:** cancer disease, lncRNAs, therapeutic drugs

## Abstract

Deep whole genome and transcriptome sequencing have highlighted the importance of an emerging class of non-coding RNA longer than 200 nucleotides (i.e., long non-coding RNAs (lncRNAs)) that are involved in multiple cellular processes such as cell differentiation, embryonic development, and tissue homeostasis. Cancer is a prime example derived from a loss of homeostasis, primarily caused by genetic alterations both in the genomic and epigenetic landscape, which results in deregulation of the gene networks. Deregulation of the expression of many lncRNAs in samples, tissues or patients has been pointed out as a molecular regulator in carcinogenesis, with them acting as oncogenes or tumor suppressor genes. Herein, we summarize the distinct molecular regulatory mechanisms described in literature in which lncRNAs modulate carcinogenesis, emphasizing epigenetic and genetic alterations in particular. Furthermore, we also reviewed the current strategies used to block lncRNA oncogenic functions and their usefulness as potential therapeutic targets in several carcinomas.

## 1. Introduction

Over recent years, advances in genomics have led to the discovery that the genome is far more pervasively transcribed than was previously appreciated. Much of the newly discovered transcriptome appears to represent long non-coding RNAs (lncRNAs), a heterogeneous group of largely uncharacterized transcripts [1,2,3]. These lncRNAs share many features with protein-coding RNA transcripts, such as the presence of epigenetic marks indicating differential expression [4], the presence of introns, transcription mediated by RNA polymerase II or, in a few cases, by RNA polymerase III and the existence of spliced variants. Many but not all lncRNAs are polyadenylated, and there is also evidence indicating that many lncRNAs exist in both polyadenylated and non-polyadenylated forms [5]. LncRNAs can overlap coding genes, and indeed, it is estimated that 20% of human transcripts display sense-antisense pairs [6]. These transcripts may overlap the entire gene or only a part of it, and a non-coding transcript may originate from either the sense or antisense strands [7].

Although the expression levels of lncRNAs are very low compared with the protein coding RNAs, lncRNAs exhibit a much more restricted temporal and tissular expression [8]. Such specificity is key in the role played by a multitude of lncRNAs in certain biological processes. At the cellular level, lncRNAs can be located in both the cytoplasm and the nucleus and can even be found in both subcellular locations. Indeed, its subcellular location is a reflection of the functional role of these lncRNAs within the cell [9]. Thus, cytoplasmic lncRNAs have mainly a regulatory function at the post-transcriptional level, while nuclear lncRNAs have mainly a regulatory function at the transcriptional level, although there are examples of nuclear lncRNAs that can be exported to the cytoplasm and thus exert regulatory functions at the post-transcriptional level [10,11,12].

In brief, lncRNAs have most commonly been classified into six well-established groups based on their genomic location and the genetic elements that surround them. The first is lncRNAs transcribed from the same promoter as the gene adjacent to them. This type of lncRNA can be transcribed in the 3′ or 5′ directions and can be transcribed from the same DNA strand or from the complementary strand. Both the expression of the lncRNA and the messenger RNA of the adjacent gene are usually correlated, with the expression of the messenger RNA being modulated by the action of the lncRNA. The second group is lncRNAs located between two protein-coding genes at a distance of approximately 10 kb between them in so-called genomic deserts. These lncRNAs are denominated as intergenic long non-coding RNAs (lincRNAs) and constitute the main class of lncRNAs in the genome [13]. The third group is lncRNAs transcribed from introns of genes that code for proteins or from promoter regions (ilncRNAs). There is a subclass of intronic lncRNAs called snoRNAs. This subclass does not have the typical structure of other lncRNAs and has exclusively a nuclear location [14]. The fourth group is lncRNAs derived from promoter regions located within active promoters that are transcribed (eRNAs) [15,16]. The fifth group is circular lncRNAs (circRNAs) that can be generated during alternative splicing of protein-encoding genes [17]. Finally, there is another class of lncRNAs that contains microRNAs within their genetic structures, as exemplified by H19, whose first exon encodes miR-675 [18].

The Cancer Genome Atlas Consortium project, together with other large-scale sequencing projects aimed at characterizing cancer genomes as well as the possible epigenetic and genetic dysregulations that configure them, have provided a precise molecular characterization of approximately 11,000 primary cancers, discovering a substantial fraction of undescribed somatic abnormalities (e.g., point mutations, genetic rearrangements, and copy number alterations) [19,20,21,22]. Khuruna et al. (2013) reported that 99% of somatic SNVs in different carcinomas occur in non-coding regions (ncRNAs, pseudogenes, and transcription factor binding sites) [23]. Furthermore, a study based on TCGA and lncRNA expression data from TANRIC shows that mutational frequencies in lncRNAs, whose expression is affected by somatic alterations (MutLncs), are low, and that to some extent, their alteration tends to be specific to the type of disease [24].

Interestingly, numerous studies revealed a previously uncovered role for lncRNAs as conditional and constitutive oncogenes or tumor suppressor genes through their facility to regulate each and every characteristic of cancer, such as aberrant proliferation, cellular invasion, altered lipid metabolism, metastasis, and immune escape. Furthermore, regulation exerted by lncRNAs can be carried out both at the transcriptional level - epigenetic and genetic - or at the post-transcriptional level [25,26,27,28,29,30]. These findings, and especially the cancer-specific expression of most of them, pointed to lncRNAs as possible biomarkers or therapeutic targets. In this manuscript, we summarize the main mechanisms by which lncRNAs modulate different carcinomas and the current strategies used to downregulate the oncogene lncRNA expression involved in carcinomas.

## 2. Genetic and Epigenetic Contributions of lncRNA Dysregulation in Cancer

The nuclear distribution of lcnRNAs and their interaction with a several nuclear elements, such as transcription factors, chromatin remodeling complexes, or even with a DNA establishing a DNA-RNA complex, have revealed their importance in the regulation of both gene transcription and its epigenetic landscape [31,32,33]. Most types of cancer require numerous changes in nuclear transcription homeostasis in order to escape cellular control systems. These changes translate into upregulation and downregulation of a multitude of genes, oncogenes, and tumor suppressor genes by different mechanisms that result in tumor initiation, progression, and metastasis [34]. In line with this, many studies have pointed out lncRNAs as pivotal modulators of nuclear transcription, altering both the transcriptional and epigenetic cellular context responsible for increasing or repressing tumorogenesis as described below (Figure 1).

Cancer’s underlying epigenetics has been widely studied, given its pivotal role in the initiation, progression, and metastasis of most types of cancer, as well as its potential as a target for different gene therapies [35,36,37]. Epigenetics can be defined as the set of modifications that alters the structure or state of chromatin, differentially regulating its gene expression without changes in the DNA sequence [38]. Epigenetic alterations can be included in two basic groups. The first is modifications in the DNA, basically produced by the addition of methyl groups at the 5′ end of certain cytosines located in the so-called CpG islands, where long stretches of CG dinucleotides harbor the promoter regions of the genome that present high, precise, and intense gene regulation [39]. Methylation of the cytosines prevents binding of the transcription factors to the promoter regions, which results in gene repression. Interestingly, many types of cancer display a specific pattern of hyper-methylated CpG islands in several tumor suppressor gene promotors, inhibiting their expression and leading to an increase in different subpopulation cancer cells and thus enhanced tumor development [40,41,42]. The second group is modifications in the histone tail, essentially caused by methylation, acetylation, or phosphorylation of certain lysines located in the N-terminal region of histones 3′, altering the nucleosome charge, affecting the chromatin state, and thus allowing or preventing the recruitment of transcriptional co-activators to it [43]. These modifications are carried out by chromatin remodeling complexes that add open or close marks to the genome, making it accessible to the transcriptional machinery [44]. Unlike the hyper-methylation of the CpG islands, in many types of cancer, hypo-methylation of the genome is observed, leading to ectopic activation of specific oncogenes, whose expression is inhibited in homeostatic conditions. Ectopic activation of these oncogenes is essential for the cell to acquire an oncogenic phenotype and escape from cellular control systems [45,46]. In the same way, some of these chromatin remodeling complexes show enhanced or reduced activity in malignant cells, which leads to dysregulation in the chromatic structure, promoting the expression of genes that enhance tumor development [47,48,49].

Emerging studies about the role of lncRNAs in carcinogenesis have pointed out as complex regulators in the epigenetics of cancer. As a consequence, many reports of cancer-related lncRNAs have been described as pivotal modulators of the epigenetic landscape by interaction with chromatin at four levels, which are described below: (1) lncRNAs as modulators of histone methylation; (2) lncRNAs as modulators of histone acetylation; (3) lncRNAs as modulators of DNA methylation; (4) lncRNAs as post-transcriptional regulators of the epigenetic apparatus and (5) (Figure 2).

### 2.1. lncRNAs as Modulators of Histone Methylation

The interaction between the chromatin remodeling complex PRC2 and HOTAIR was the first reported example of an lncRNA involved in the epigenetic regulation of chromatin [50]. In breast cancer, HOTAIR is necessary to recruit PRC2 to its genomic targets, which is required for the latter to establish the repressor marks H3K27me3 in certain tumor suppressor genes that play pivotal roles in the inhibition of metastasis such as Hoxd10, PGR (progesterone receptor), PCDH (Protocadherin 10) or Jam2 (junctional adhesion molecule). Furthermore, a loss of HOTAIR function is sufficient to prevent cell invasion and metastasis in breast cancer, reflecting the role of this lncRNA as a potent oncogene and pointing it out as a possible therapeutic target [51].

PRC2 subunits can interact with different nuclear proteins or transcription factors, forming protein complexes that establish epigenetic marks on their genomic targets [52]. Tug1, an upregulated lncRNA in different gliomas, acts as a scaffold molecule necessary for PRC2 to interact with YPP1, a neuronal transcription factor. Such a new complex can increase the expression of different genes involved in neuronal differentiation, such as BDNF, NGF, or NFT3, maintaining the pluripotency of malignant cells and thus increasing the aggressiveness of the glioma. Interestingly, the expression of Tug1 is detected in the cytoplasm too, where it acts as a sponge for miR-142, protecting SOX2 and MYC from degradation [53].

Maintenance of the epigenetic marks induced by PRC2 requires the participation of another complex: PRC1 [54]. ANRIL, a long non-coding RNA located in a genetic desert with a high prevalence of SNPs [55,56,57] (to date, many of these SNPs have been related to a higher prevalence of cancer), interacts with both PRC1 through the CBX7 subunit and PRC2 through the SUZ12 subunit to repress the expression of the CDKN2A/B locus and thus maintaining its silence. This locus is downregulated in many types of cancer, such as leukemia, breast cancer, pancreatic cancer, ovarian cancer, and gliomas [58,59,60,61,62]. Repression of the CDNK2A/B locus by ANRIL increases the cell proliferation of malignant cells, promoting progression and metastasis. On the contrary, ANRIL depletion reduces cell proliferation and decides the balance toward cell death, therefore reducing the risk of metastasis, pointing to ANRIL as a potent therapeutic target, especially in leukemia and prostate cancer, where it is upregulated [63,64].

HOTTIP provides another example of lncRNA upregulated in several types of cancer, such as hepatocellular carcinoma, gastric cancer, colorectal cancer, pancreatic cancer, lung cancer, prostate cancer, and osteosarcoma [65,66,67,68,69,70,71]. HOTTIP interacts with WDR5, inducing the chromatic opening through the WDR5/MTT complex. This complex upregulates the expression of the HOXA locus through the H3K4me3 marks. In turn, HOXA locus genes repress the expression of several tumor suppressor genes. Additionally, the detection of HOTTIP in the exosome samples of patients has been recognized as a possible prognosis marker in colorectal cancer [72].

Unlike the lncRNAs described above, MEG3 represents an intergenetic lncRNA associated with chromatin repressive marks that acts as a potent tumor suppressor gene. MEG3 physically interacts with the EZH2 subunit of the PRC2 complex to repress the expression of several genes involved in TFG-β signaling (e.g., SMAD2 or TGFBR1), increasing the aggressiveness of cancer by promoting invasion and metastasis. This repression is mediated by the formation of a triplex DNA-RNA complex in the GA-rich regions of a gene’s silenced promoters [73].

### 2.2. lncRNAs as Modulators of Histone Acetylation

Few cases of lncRNAs involved in histone acetylation have been described to date. Wan et al. (2013) described a pivotal role of lncRNA JADE in the earliest steps of DNA damage response (DDR) mechanisms by the modulation of acetylation machinery. As suggested in basic and preclinical studies, DDR is one of the primary anti-cancer barriers during tumorigenesis and is under complex and tight regulation, including alteration of the acetylation patterns of numerous gene promoters [74,75,76]. Clinical studies performed in breast cancer patients have shown lncRNA JADE upregulated expression compared with controls. Mechanistically, lncRNA JADE is required for correct JADE1 expression, a protein necessary to determinate the histone H4 substrate specificity of HBO1, which in turn mediates histone H3–H4 acetylation [77]. Several types of cancer display upregulation of HBO1, positively modulating the expression of proliferation-promoting genes linked to a poor cancer prognosis [78,79]. The depletion of lncRNA JADE is translated in the reduced growth of breast cancer in vivo, as well as in HBO1 deficient cells, suggesting a potential effect as inductors of the proliferation of maligned cells [74].

Another case of histone acetylation mediated by lncRNA was reported for lncPRESS1, which plays a pivotal role in the switching of the pluripotent or differentiated state of embryonic stem cells (ESCs) by acting as a decoy molecule from SIRT6. LncPRESS1 competes with SIRT6 for their genomic targets, avoiding that this enzyme can de-acetylate and thus the active gene expression required for promoting cell differentiation [80].

### 2.3. lncRNAs as Modulators of DNA Methylation

The patterns of aberrant methylation of lncRNA promoters have been described in many types of cancer, pointing out their importance in the epigenetic control of carcinogenesis [81]. However, only a few cases of lncRNAs involved in DNA methylation have been deeply studied, and their functions remain to be fully elucidated. TARID1 is an intergenic lncRNA whose promoter region is located within the third CpG island of the TFC21 promoter, a known tumor suppressor gene. TARID1 binds GADD45A, a DNA repair protein that promotes active demethylation in numerous promoters. The TARID1-GADD45A complex directs it, together with TDG, a protein necessary for GADD45A to interact with its genomic targets, to the TFC21 promoter, where it eliminates methylation and allows for the expression of TFC21, which in turn plays a key role in protection against head and neck squamous cell carcinoma (HNSCC). Interestingly, TARID1 formed an R-loop with the TCF21 promoter, which was recognized by GADD45A as a region marked for demethylation [82,83].

Merry et al. (2015) uncovered a total of 148 lncRNAs that are associated with DNMT1 in colon cancer cells through RIP-seq. Among them, one lncRNA was highlighted, named DNMT1-associated colon cancer repressed lncRNA 1 (DACOR1), which is highly and specifically expressed in normal colon tissue while it is repressed in colon cancer cell lines. Furthermore, overexpression of DACOR1 in colon cancer cells resulted in a gain in DNA methylation at multiple loci without changing the DNMT1 expression level. Interestingly, DNTM1 is an important repressor of tumorigenesis [84]. ChIRP-seq analysis demonstrated that DACOR1 occupies a total of 338 genomic loci, of which 161 peaks are near 150 annotated genes. Remarkably, 31 of these sites overlapped with differentially methylated regions previously found in colon cancer samples with respect to normal tissues. These findings indicate that DACOR1, cooperating with both chromatin and DNMT1, targets the DNMT1 protein complex toward exact genomic loci. Furthermore, DACOR1 was found to repress the expression of cystathionine β-synthase (CBS) and, in turn, increase methionine, which is the substrate to produce S-adenosyl methionine (SAM). SAM is a necessary methyl donor for DNA methylation in mammalian cells. Thus, DACOR1 may also impinge on DNA methylation through orchestrating cellular SAM levels [85,86].

### 2.4. lncRNAs as Post-Transcriptional Regulators of Related Epigenetic Proteins

The interaction between lncRNAs and the related epigenetic protein landscape is not limited to recruitment, scaffold, or active element functions required for chromatin remodeling. Furthermore, lncRNAs have been reported as pivotal players in modulating the chromatin protein complex stability by promoting protein degradation, exerting protectives roles in most of cases to trigger pro-oncogenic epigenetic marks or inhibit protein degradation, acting as oncogenes. For example, ANCR is capable of directly binding to the EZH2 subunit, promoting its degradation. ANCR-EZH2 binding is required for CDK1 to target EZH2 for ubiquitin–proteasome degradation via Thr-345 and Thr-487 phosphorylation in breast cancer cells. Curiously, in breast cancer, ANCR expression is inactive, leading to hyperactivity of EZH2, which in turn sets up several repressive marks in tumor suppressor genes such as Hoxa10 or E-Cadherin, which are involved in progression and EMT signals [87]. Note that the modulation of ANCR has been probed in other carcinomas [88,89].

EZH2 degradation is not modulated solely by ANCR. MEG3 promotes EZH2 ubiquitination, leading to upregulation of LATS2, a tumor suppressor kinase that inhibits cell proliferation and metastasis through the Hippo signaling pathway in several types of cancer [90]. Gallbladder cancer also displayed a downregulation of MEG3 accompanied by LAST2 downregulation and thus increased cell proliferation and metastasis [91].

Unlike ANCR or MEG3, LUCAT1 is considered an oncogene by protecting the protein degradation of DNMT1 in esophageal squamous cell carcinoma. A depletion in LUCAT1 expression is correlated with reduced levels of DNMT1 expression together with the upregulation of UHRF1, a protein involved in ubiquitination and therefore DNTM1 degradation [92].

### 2.5. lncRNAs as Nuclear Environment Modulators

The nuclear compartment not only encloses the chromatin in its different phases and the nucleolus but also contains different structures of mostly irregular shapes, known as nuclear bodies [93]. These structures exert pivotal roles in the transcriptional regulation of several pathways related to distinct cellular processes, such as the differentiation, proliferation, or maintenance of homeostasis and, therefore, disease development too [94,95,96,97]. Many reports have highlighted the role of these structures in cancer pathogenesis, involving them in the underlying transcriptional regulation of tumorigenesis [98,99,100]. Along different nuclear bodies, paraspeckles, substructures located on the periphery of the nucleus, were first discovered in 2002 [101] and have emerged as pivotal regulators of nuclear function modulating. First, they distribute nucleus proteins and are available to interact with chromatin or transcriptional machinery [102]. Second, they retain mRNAs, avoiding their transport to the cytoplasm and thereby translation. Interestingly, mRNAs retained by paraspeckles are exported later to the cytoplasm, considerably increasing the number of messenger RNA molecules and their translation. However, the underlying biological processes that determine the export time lapse are poorly understood [103,104]. Third and finally, they sequester proteins related to miRNA biogenesis and processing [105]. The formation of paraspeckles requires the presence of NEAT1, or long non-coding RNA nuclear paraspeckle assembly transcript 1 [106,107]. NEAT1 is transcribed in two isoforms—Neat1.1 and Neat1.2—displaying different RNA processing, which results in the generation of two different transcripts both in their length and in their structural motifs. While the first isoform is dispensable in paraspeckle biogenesis [108], the second isoform is responsible for paraspeckle assembly, constituting a limiting factor for the formation of these nuclear bodies and thus determining the tendency of the nucleus to form them [109]. Zeng et al. (2018) reported the pivotal importance of Neat1.2 and therein paraspeckles as promotors of the aggressiveness of Chronic myeloid leukemia (CML). SPKQ, a bivalent protein that can act as a splicing factor required in paraspeckle formation or as a transcription factor exerting the activation of apoptotic proteins such as BLC2, BBC3, or BAX, is associated with Neat1.2 through the C motif. Neat1.2-SPKQ binding reduces the availability of this protein to act as a transcription factor, thus reducing the expression of apoptosis factors BLC2, BBC3, and BAX and leading the cell to escape apoptosis, thereby enhancing the growth and proliferation of CML. Curiously, Neat1.2 expression is downregulated by c-Myc, a known repressor of CML progression. Neat1.2 repression mediated by c-Myc reduces paraspeckle formation and leads to SPKQ binding to the promoters of the apoptotic genes referred to above, activating their expression and consequently promoting apoptosis and achieving a better prognosis [102].

## 3. Transcriptional Gene Modulation by lncRNAs

The 3D architecture of the genome is essential for gene transcriptional regulation. RNA polymerases require contact with the promoter regions of the genes to be transcribed [110]. Contact between the distal regions of the genome is facilitated by the transcription factors and active enhancer regions, which generate chromosomal looping. Altering the 3D structure of the chromatin allows RNA polymerases to recognize the distal promoter regions and initiate the synthesis of different transcripts [111]. Active enhancers are indispensable for leading transcription genes that are located far from each other in the genome at the same time [112,113,114]. Under homeostatic conditions, active enhancers control the maintenance of different cell types. However, they are deregulated in many human cancers. Along with them, lncRNAs derived from the transcription of active enhancers are differentially expressed in cancer tissues and have been revealed to act as oncogenes, promoting the transcriptional activation of the oncogenic pathways and even inducing chromosomal rearrangements and genomic instability [115,116,117,118].

The regulatory impact of enhancer-related lncRNAs has been elucidated, especially in human T cell acute lymphoblastic leukemia (T-ALL). Leukemia-induced non-coding activator RNA-1 (LUNAR1) was recognized in an integrated transcriptome profile from T-ALL patients as a specific lncRNA involved in cell growth both in vitro and in vivo in the early stages. LUNAR1 is upregulated by the Notch1/Rbp-jk activator complex, which plays a pivotal role in the initiation of T-ALL carcinogenesis [119,120]. LUNAR1 expression is necessary to promote the IGF1R mRNA levels and to maintain the IGF1 signaling required to stimulate tumor growth. The depletion of LUNAR1 leads to downregulation of IGF1R and a reduction of both the Mediator complex and RNA Pol II binding to both the IGF1R enhancer and promoter [119].

Recently, Tan et al. (2019) described ARIEL, an ARID5B-inducing enhancer associated long non-coding RNA - another example of eRNA involved in T-ALL pathogenesis. The expression of ARIEL is associated with the ARID5B enhancer activity and is required for the recruitment of the Mediator complex toward the ARID5B promoter and thereby increasing the expression of this transcription factor. ARID5B is necessary to activate the TAL1-induced transcriptional program and the MYC oncogene [121]. Curiously, TAL1 positively modulates ARIEL expression, showing a feedback regulatory system. While ARIEL knockdown in cells is translated into growth inhibition, murine model mutants to ARIEL display a block in disease progression, reflecting the importance of this lncRNA in T-ALL pathogenesis and pointing out a possible preventive therapeutic target [122].

## 4. Post-Transcriptional Gene Regulation by lncRNAs Involved in Cancer

Post-transcriptional gene regulation includes the modulation of mRNAs, promoting or inhibiting their stability, interaction with ribosomal machinery facilitating or blocking the protein translation process, and finally affecting alternative splicing to generate different transcripts. Although epigenetic and genetic regulation have historically been considered the pivotal axis for the development of most carcinomas, emerging evidence has shown that post-transcriptional regulation of several genes involved in cancer diseases also exerts a key role [123,124].

The functional complexity of lncRNAs, together with their dynamic cellular localization, allows them to exert post-transcriptional gene regulation at different levels within both the cytoplasm and the nucleus [125,126] as follows: (1) competing by binding to microRNAs and thus acting as a sponge that eventually prevents triggering their targets; (2) promoting or inhibiting the translation of mRNA; (3) interacting with ribosomal machinery, modulating the biogenesis, translocation, and binding of ribosomal subunits; and (4) interfering in the splicing process through the modulation of different splicesome proteins [127,128] (Figure 3).

### 4.1. lncRNAs Acting as Competing Endogenous RNAs

Many cytoplasm-located lncRNAs exert themselves as master regulators of post-transcriptional gene modulation by competing with protein coding RNA 3′UTRs and thus binding to microRNAs [129,130]. Mechanistically, competing endogenous features reside in the similarity between the sequence of a certain lncRNA and the 3′UTR region of an mRNA containing repetitive binding sites for a determined microRNA. The lncRNA-microRNA binding complex prevents it from being loaded into the AGO2/RISC complex, and therefore, microRNA is incapable of physically interacting with its mRNA targets.

Phosphatase and tension homolog pseudogene 1 (PTENP1) was the first provided example of ceRNA involved in carcinogenesis [131]. PTENP1 displays high homology with the PTEN gene, preserving its sequence-binding sites for the different microRNAs that recognize the 3′UTR region of PTEN (miR20a, miR19b, miR21, miR26a, and miR214). The expression of both genes is downregulated in different carcinomas, although PTENP1 expression is not dependent to PTEN. Knockdown of both genes is translated into growth acceleration, while upregulation of PTENP1 is enough to promote growth suppression in prostate cancer. To date, many reports have pointed out that the inhibitory role of PTENP1-PTEN genes is not exclusive to prostate cancer, and it is extensible to several type cancers such as colorectal cancer, breast cancer, or oral squamous cell carcinoma [132,133,134]. Thus, PTENP1 is considered a master tumor suppressor gene, as demonstrated by the inhibition of functional conservation between different carcinomas.

In sharp contrast, HULC provides an example of the oncogenic sponge long non-coding RNA involved in several carcinomas, promoting tumor angiogenesis in liver cancer and aberrant cell proliferation in leukemia and impairing lipid metabolism in hepatoma cancer [135,136,137,138,139]. For example, sequestering miR-107 by HULC in liver cancer is indispensable for promoting angiogenesis and increasing the aggressiveness of maligned cells. The miR-107 targeting E2F1 mRNA, a pivotal protective factor, promotes angiogenesis by inhibiting SPHK1, a specific kinase upregulated in liver cancer [140]. HULC is not an exclusive example of an lncRNA capable of recognizing and binding to miR107. Neat1, with nuclear functions that were described above, also exerts a pivotal role as an oncogene in both laryngeal squamous cell cancer and glioma, sequestering miR107 and repressing the miR107/Cdk6 pathway, which plays a protective role in inducing apoptosis and cell cycle arrest at the G1 phase [141,142]. Furthermore, a large subset of miRNAs is recognized and targeted by Neat1 (which, in all cases reported, acted as a potent oncogene) in several tumors, such as miR-214 in endometrial and thyroid carcinoma [143,144], miR-101, miR-218, and mR-211 in breast cancer, or miR-506 in pancreatic carcinoma [145,146,147,148]. Therefore, Neat1 binding to sponge miRNAs is translated into the promotion of tumor cell growth, proliferation, migration, invasion, metastasis, and maintaining a stem cell-like phenotype.

### 4.2. lncRNAs Promoting or Inhibiting the Translation of mRNA-Targets

Many lncRNAs modulate mRNA expression, promoting or repressing the translation of specific mRNA targets that interact directly with them. Analysis of the transcriptome of human cervical carcinoma (HeLa) cells has clarified a post-transcriptional role of lncRNA-p21, which is located in the cytoplasm in a close relationship with the ribosomes. lincRNA-p21 interacts with the mRNA of two apoptotic cell-protective proteins—CTNNB1 and JUNB—both of which are upregulated by the HuR gene [149,150,151]. Interestingly, lncRNA-p21 expression is inhibited by HuR, a known inducer of cell migration and tumor aggressiveness in HeLa cells [152,153]. The RNA-RNA complex formed by the mRNA of lncRNA-p21 and the CTNNB1 and JUNB 3′UTR regions triggers the association of these mRNAs with the translational repressors RCK and FMRP, which leads to the attenuation of their translation [151,153].

### 4.3. lncRNAs Interact with Ribosome Machinery, Modulating the Biogenesis, Translocation, and Binding of Ribosomal Subunits

Few lncRNAs have been described as modulators of protein biogenesis at the ribosome level, affecting biogenesis, translocation, or ribosomal subunit formation. SLERT is a clear example of lncRNA being involved in the regulation of ribosome biogenesis. SLERT (Sno-RNA-terminated lncRNA enhances pre-ribosomal RNA transcription) acts as an oncogene in several human cancer cell lines that positively modulates pre-rRNA transcription [154]. Tumor development requires increased protein biogenesis and, consequently, increased ribosome biogenesis to meet the demand for de novo protein synthesis from carcinoma cells [155]. Mechanically, SLERT interacts directly with DEAD-box RNA helicase DDX21 and prevents it from coating polymerase I and thus reducing the transcription of ribosomal RNAs. Furthermore, the binding of SLERT and DDX21 allows RNA polymerase I to transcribe pre-rRNAs. Consistent with this, the inhibition of SLERT reduces the tumorigenic potential both in vitro and in vivo, showing a positive role for this lncRNA in carcinogenesis [154,156].

### 4.4. lncRNAs Involved in Splicing Alternative Mechanisms

Alternative splicing is a complex regulatory system that allows the generation of different RNA transcripts from the same gene. The modulation of alternative splicing by lncRNA molecules is still unclear, but some cases have been reported. ZEB2-NAT was described first as an antisense transcript of ZEB2, a pivotal gene involved in reducing EMT by repressing E-Cadherin [157]. Curiously, ZEB2 shows an IRES sequence harboring an intron located upstream between the first and second exon. The IRES sequence plays an essential role in recruiting ribosomes, initiation factors, and RNA-binding proteins. E-cadherin repression by ZEB2 requires such an IRES sequence. Retention of the IRES sequence in the mature mRNA of ZEB2 is mediated by the activity of ZEB2-NAT, which interacts directly with the IRES sequence and avoids splicing processing. The downregulation of E-Cadherin by ZEB2-NAT points it out as a possible oncogene, since it promotes the maintenance and metastasis of several human carcinomas [158].

## 5. Development of Strategies to Obtain the Blocking Oncogene Functions of lncRNAs

Theragnostics is the nexus point of therapeutics and diagnostics and describes a system to customize healthcare using molecular and genetic tools for treatment decisions tailored to an individual patient [159]. The emerging functions of many lncRNAs as oncogenes in several carcinomas promoting different steps in carcinogenesis, such as escape to control cellular systems, cell proliferation and migration, aberrant and enhanced metabolism, impaired and deregulated epigenetic landscape, and metastasis, have pointed out lncRNAs as potential therapeutic targets in the breaking of tumorigenesis [160,161]. The importance of lncRNAs in each and every one of these processes has led to the design of different strategies and molecules that allow repressing the function or expression of the lncRNAs, since in most of the examples described, blocking the lncRNA function would be translated in repression of the aggressiveness of cancer accompanied by the reduction of several promoting processes related to carcinogenesis [162]. Thus, lncRNAs are pinpointed as having the perfect theragnostics to design personalized medicines based upon the molecular features of each tumor. Currently, against lncRNAs with direct or indirect oncogenic function, several targeting strategies have been designed by modulation at the (1) genome-level, repressing the expression levels of lncRNAs or oncogene targets by a deletion gene, and (2) post-transcriptional RNA-level, affecting the stability of the lncRNA molecules and triggering degradation.

### 5.1. Genomic Modulation of lncRNAs by a CRISPR-Based System Edition

The intricate genomic architecture of lncRNAs, together with the limited knowledge of the biology of their promoters and the underlying transcriptional context, limits the use of a type II CRISPR-system, a classical and basic CRISPR-based system that was briefly widespread in genetic edition assays. Curiously, Goyal et al. (2017) identified that around 62% of the total lncRNAs can be classified as “non-CRISPable” because of the presence of internal or bidirectional promoters in their sequence [163]. Furthermore, in many cases, bidirectional promoter lncRNAs are related with the transcription of neighboring genes and the correct expression of them. Currently, several CRISPR system modifications have been performed for the knockdown expression of lncRNAs, including the double-excision CRISPR knockout (DECKO) system and a tandem of the CRISPR interference (CRISPRi) and “dead”-Cas9 (dCas9) systems [164,165,166]. The efficiency of both in the downregulated expression of lncRNAs has been probed for more than 500 lncRNAs involved in cancer diseases [167,168,169]

### 5.2. Post-Transcriptional Modulation of lncRNAs by Inhibitory Molecules

Antisense oligonucleotides (ASOs) are single-strained antisense oligonucleotides containing a central stretch of deoxyribonucleotides flanked on both sides by RNA nucleotides. The DNA stretch is often phosphorothioated to improve the nuclear stability of the ASO, and chemical modifications are added into the nucleotides to enhance the efficiency activity and promote nuclear translocation. At the nucleus, the DNA contained in the ASO sequence recognizes a small fragment of a complementary lncRNA DNA-RNA heteroduplex, which is then cleaved by nuclear endogenous RNAase H1, triggering degradation and thus a reduction in lncRNA expression [170,171]. ASOs are widely used to repress the expression of nuclear lncRNAs or cytoplasmic lncRNAs that exert their functions at the nuclear level [172,173,174,175]. Different modifications of a basic sequence of ASOs have provided additional types of repressive molecules with different specifications, among which it is important to highlight LNA GapmeRs, antagoNATs and mixmers. First of all, they are very similar to ASOs, with the exception of adding LNA to the flanking arm to improve the binding affinity and nuclease resistance [176]. LNA GapmeRs function as efficient regulators to target epigenetic modifications in vivo for therapeutic applications, as has been demonstrated [177]. AntagoNATs are short, single-stranded oligonucleotides that display a high homology to specific antisense lncRNA. The complex formed by AntagoNATs and lncRNA avoids the latter being recognized as their targets. Modarresi et al. (2012) achieved downregulation of BDNF-AS1 using a specific AntagoNAT with LNA substitutions at each end and phosphorothioate-modified backbones. BDNF-AS1 repression upregulates BNDF expression and activates the brain-derived neurotrophic factor, which is a protective modulator in several gliomas. Unlike LNA GapmeRs and AntagoNATs, the repression exerted by mixmers is not mediated by RNAaseH1 [178].

Small interfering RNAs (siRNAs) are short double-stranded RNAs that, once inside a cell, are cleaved into a single-stranded RNA capable of creating a base pair complementary for an lncRNA of interest. The RNA-lncRNA base pairing is recognized by the RNA-induced silencing complex (RISC) that follows its Argonautic degradation [179]. Many types of siRNAs have been probed in cancer treatment to silence lncRNAs trying to attain better knockdown expression. Although several reports have exemplified the repression of cancer-related lncRNAs using classical siRNA, it represents a temporal limitation, since the treatment effect disappears after 24–72 h, depending on the lncRNA cellular expression and location. For example, Prensner et al. (2013) reduced the cell proliferation and invasiveness of several prostate cell lines using different siRNAs against SChLAP1, a known prostate-related lncRNA that promotes aggressive prostate cancer through the inhibition of the tumor-suppressive activity of the SWItch/Sucrose Non-Fermentable (SWI/SNF) complex [180]. A more stable and efficient form of siRNA is provided by short hairpin RNA (shRNA). This type of siRNA is capable of DNA integration and consists of two complementary 19–22 bp RNA sequences linked by a short loop of 4–11 nt to the hairpin found in the miRNA. Following transcription, the shRNA sequence is exported to the cytosol, where it is recognized by an endogenous enzyme Dicer and processed, generating double-strand RNA duplexes [181]. shRNAs are widely used for both in vitro and in vivo treatment. For example, Sun et al. (2016), using a Malat-shRNA vector, were capable of preventing metastasis and invasion in vitro and in vivo in cervical cancer. Malat-shRNA vector treatment was translated into the downregulation of Malat1 and, therein, downstream targets such as β-catenin, Snail, or vimentin, markers of EMT in cancer cells [182]. Curiously, shRNA vectors not only can contain a sequence of lncRNA that is of interest, but they can also harbor sequences that recognize other oncogenic factors, generating more than one shRNA within the cell and resulting in a more efficient reduction of the cancer phenotype. An example of this was provided by Zong et al. (2019) in the treatment of glioma carcinoma. In this type of tumor, LMX1A activates NLRC5 expression, stimulating the Wnt/β-catenin signaling pathway, which promotes malignancy of the glioma cells. LMX1A expression is repressed by protector miR-499, which recognizes 3′UTR LMX1A and triggers its mRNA degradation. Furthermore, glioma cells exhibit upregulation of SCAMP1, a sponge lncRNA that binds to miR-499, avoiding LMX1A repression. The downregulation of SCAMP1 is translated into lower cell proliferation, migration, and invasion and enhanced apoptosis, suggesting that sequestering miR-499 via SCAMP1 exerts a critical role in the pathogenesis of glioma carcinoma [183].

### 5.3. Small Molecules against lncRNAs as Therapeutic Drugs

Currently, searching for small molecules that can bind to lncRNAs and block their action is the main goal in the design of effective and pioneering treatment of several cancers. Exploring the type of approach in order to find out the possible blocking molecules of Malat oncogene functions in breast cancer, Fardokht et al. (2019) carried out a small molecule microarray strategy, obtaining two different ligands with the capacity to specifically recognize the ENE triplex of Malat1, which protects them from degradation, leading to the accumulation of high levels of Malat1 transcripts in the nucleus. Ligand administration is translated into the downregulation of Malat1 expression both in the cell lines of breast cancer and the branching morphogenesis in a mammary tumor model by inducing structural changes. Curiously, the structure of one ligand is similar the Neat1 ENE triplex too. However, despite sharing a high degree of homology, it is not capable of interacting with Neat1 or reducing its expression levels, showing that the design of these repressor ligands is specific to a single lncRNA [184].

## 6. Conclusions and Future Perspectives

The study of the human transcriptome has shifted our understanding of gene expression and regulation. lncRNAs taking part in multiple cellular regulatory networks have revealed their importance in homeostasis, their implications in cancer, and their revolutionary effects on our perspective of the disease from its origins to the design of novel therapeutic strategies. Theoretically, the effects of lncRNA networks modulating cellular metabolism in cancer can impact the regulation of cellular metabolism and energy homeostasis. The studies, however, are still in their infancy and far from using lncRNAs in their actual state in the clinical arena, with one of the greatest challenges being identifying the sequences and structural elements that allow lncRNAs to carry out their cellular functions. lncRNAs are promising molecules for applications in therapy, especially for regulatory networks of cancer cells, but one major requirement is a better understanding of lncRNA functions and mechanisms in terms of both physiological and pathological conditions. It is necessary to concentrate our efforts on their functional study, and hence, intensive research paired with lncRNA characterization will lead to progress in understanding the lncRNA code. Finally, the structural and functional novelty of lncRNAs offers promising anti-cancer therapeutics that may avoid the emergence of drug resistance like that displayed with currently used therapies. Further studies to better understand the molecular mechanisms of lncRNAs in cancer offer the prospect of the development of more effective cancer therapies.

## Figures and Tables

**Figure 1 ijms-23-00764-f001:**
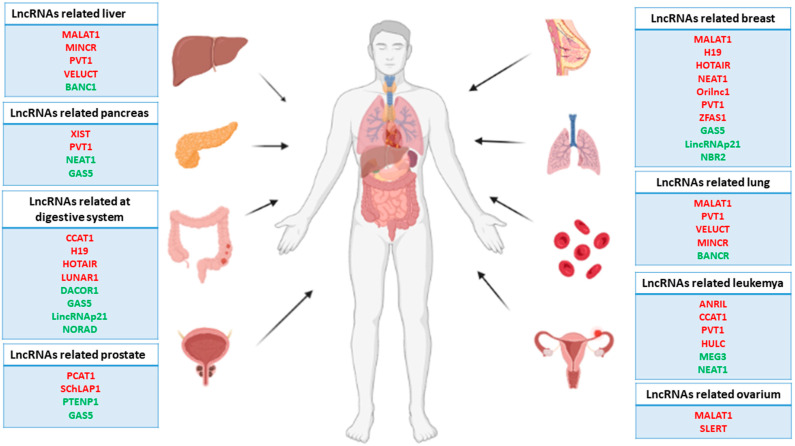
Representative scheme of the main human carcinomas and related lncRNAs. Note that the red lncRNAs act as oncogenes, while the green lncRNAs act as tumor suppressor genes.

**Figure 2 ijms-23-00764-f002:**
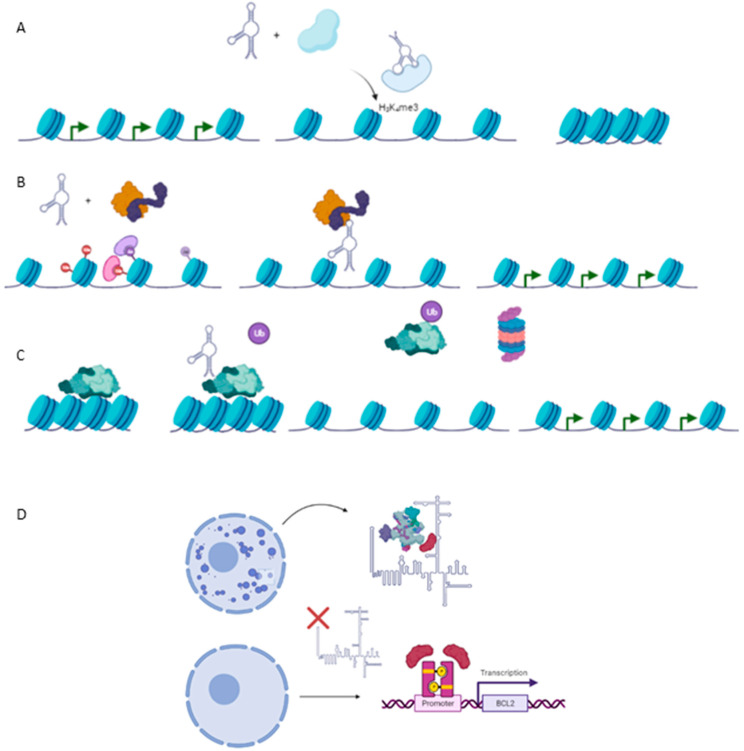
Epigenetic mechanism involved in carcinogenesis. (**A**) Histone methylation exerted by HOTAIR-PRC2 complex to repress expression of several genes such as Hoxd10, PGR, PCDH, or Jam2 with a protective role against metastasis in breast cancer cell line. (**B**) TCF21 promoter demethylation exerted by TARID1-GADD45A, which positively modulates expression of the TCF21 gene, a protective factor against head and neck squamous cell carcinoma (HNSCC). (**C**) Ubiquitination of EZH2, a subunit of PRC2, by ANCR reducing negative marks at Hoxd10 or E-Cadherin genes, exerting a protective role against EMT and metastasis in breast cancer cell line. (**D**) Required binding SPKQ-Neat1 for the formation of paraspeckles. Downregulation of Neat1 is translated in the high availability of SPKQ and therein the upregulation of apoptosis genes such as BLC2 or BAX.

**Figure 3 ijms-23-00764-f003:**
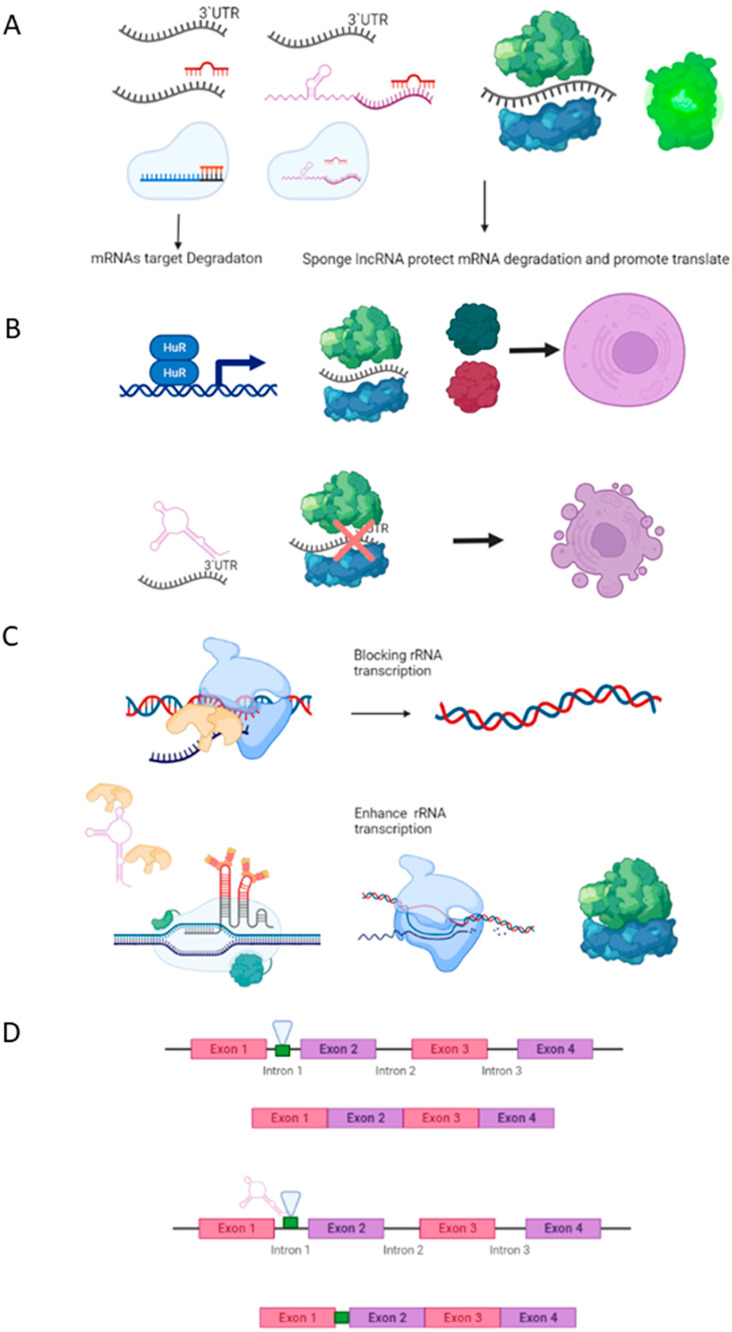
Main regulatory mechanisms of lncRNAs at the post-transcriptional level. (**A**) Competing sponge lncRNAs act to form an lncRNA-miRNA complex, avoiding the degradation of 3′UTR target genes, therein repressing the translation of them. (**B**) lncRNAs binding mRNA targets and avoiding the ribosome-initiated translation process. In sharp contrast, the mRNA-lncRNA complex attenuates the binding of repressed protein translation. (**C**) lncRNAs can act as positive modulators of rRNA synthesis, increasing the ribosome pool necessary for the increased protein demand in cancer cells. (**D**) lncRNAs can modulate the splicing process, leading to transcription of different isoforms that exert pivotal roles in several carcinomas.

## Data Availability

Not applicable.

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
