# Peer review of "Molecular Mechanisms of lncRNAs in the Dependent Regulation of Cancer and Their Potential Therapeutic Use"

_ijms, 2022, doi:10.3390/ijms23020764_

Round 1

Reviewer 1 Report

The review manuscript entitled “Molecular mechanisms of LncRNAs in cancer and their potential therapeutic use” by Padilla et.al. discussed the genetic and epigenetic impact of lncRNAs in different types of carcinogenesis. They also explained the role of lncRNAs in the regulation of different stages of gene expression ranging from transcription and posttranscriptional processing to RNA stability and translation in cancer cells. Finally, they mentioned the CRISPR and ASO based strategies to regulate the lncRNA based carcinogenesis. In brief, the manuscript nicely explains the involvement of lncRNAs in different types of cancer and possible therapeutic approaches that can be applied to regulate lncRNA dependent carcinogenesis.

Following are some suggestions that should be considered to make the manuscript more informative and interesting for a wide range of readers.   

Major Comments:

  1. A pictorial presentation of the involvement of lncRNA at different stages of chromatin maintenance or gene expression would be very helpful to clearly understand the mechanism of their action in different biological processes of cancerous cells.
  2. In the manuscript, the author discussed the role of some representative lncRNAs in carcinogenesis but some very well studied lncRNAs like NEAT1 has not been mentioned. NEAT1 is a very well-studied lncRNA that is involved in the paraspeckle formation and it regulates RNA retaining, mRNA breakage, A-to-I editing and protein seizing. It has a dramatic effect on different types of cancer cells and has altered expression in tumorous cells.
  3. The author mentioned only two possible therapeutic approaches- CRISPR and ASO to treat the lncRNA dependent carcinogenesis which seems to be incomplete information. They should provide information about other approaches like RNAi, Drugs etc that can be used in lncRNA dependent carcinogenesis.

  1. The manuscript needs a careful improvement in English style and grammar.

Specific Comments:

  1. The title of the manuscript looks incomplete. A mechanism is based on a process, not a molecule. “Molecular mechanism of lncRNAs in cancer….” could be “Molecular mechanism of lncRNAs dependent regulation of cancer…..”
  2. Does it really needed to put the subheading “2.1. Epigenetic landscape modulation by lncRNAs” lane 97. Since there is not any further such subgroup, this seems to be unnecessary.
  3. Typos

Lane 19                 “Until” must be “the”

Lane 20                 “ad” must be “and”

Lane 20                 “of types” must be “types”

Lane 96 is not clear

Lane 110 is not clear

Author Response

Reply to reviewer #1

First of all, we would like to thank the reviewer for his/her constructive comments that certainly will be helpful to improve this manuscript.

Reviewer #1.

A pictorial presentation of the involvement of lncRNA at different stages of chromatin maintenance or gene expression would be very helpful to clearly understand the mechanism of their action in different biological processes of cancerous cells.

Answer to Reviewer #1.

Following the reviewer´s recommendation, we have elaborated more detailed figures 2 and 3, including different representations of main molecular mechanisms of lncRNAs both in transcriptional and post-transcriptional regulation.

Reviewer #1.

In the manuscript, the author discussed the role of some representative lncRNAs in carcinogenesis but some very well studied lncRNAs like NEAT1 has not been mentioned. NEAT1 is a very well-studied lncRNA that is involved in the paraspeckle formation and it regulates RNA retaining, mRNA breakage, A-to-I editing and protein seizing. It has a dramatic effect on different types of cancer cells and has altered expression in tumorous cells.

Answer to Reviewer #1.

In agreement with the reviewer, we have emphasized the role of Neat1 in the formation of paraspeckles and included different reports that show the role of Neat1 as a sponge of different protective miRNAs in tumorogenesis (as indicated in the revised version. section 2.5. lncRNAs as nuclear environment modulators).

Reviewer #1.

The author mentioned only two possible therapeutic approaches- CRISPR and ASO to treat the lncRNA dependent carcinogenesis which seems to be incomplete information. They should provide information about other approaches like RNAi, Drugs etc that can be used in lncRNA dependent carcinogenesis.

Answer to Reviewer #1.

Following his/her suggestions, we have extended (as indicated in the revised version, section 5.2 and new 5.3) the information on the different therapeutic strategies of lncRNAs including siRNAs, shRNAs and small inhibitory molecules.

Reviewer #1.

The manuscript needs a careful improvement in English style and grammar.

Answer to Reviewer #1.

The manuscript has been revised, and suggestions, such as title changes, have been addressed.

Reviewer 2 Report

García-Padilla et al. have summarized the current knowledge on the role of lncRNA in the field of cancer focusing on the molecular mechanisms involved. In addition they describe the possibility to target such lncRNA as therapeutic targets.

While the manuscript is initially well written, it is apparent that more recent functions of lncRNAs are missing from the analysis, together with cancer-specific lncRNAs of which mechanisms have been described. Also, they often refer to review papers, while primary works should be cited.

Author Response

Reply to reviewer #2

First of all, we would like to thank the reviewer for his/her constructive comments that certainly will be helpful to improve this manuscript.

Reviewer #2.

While the manuscript is initially well written, it is apparent that more recent functions of lncRNAs are missing from the analysis, together with cancer-specific lncRNAs of which mechanisms have been described. Also, they often refer to review papers, while primary works should be cited.

Answer to Reviewer #2.

In agreement with the reviewer, we have included in this revised version more recent functions and mechanisms of lncRNAs. Thus,

  1. we have included the section “2.5. lncRNAs as nuclear environment modulators”;
  2. we have included the section “5.3. Small molecules against lncRNAs as therapeutic drugs”; and
  • we have improved the section “5.2. Post-transcriptional modulation of lncRNAs by inhibitory molecules”.

Following his/her suggestions, we have modified the reference list, including bibliography corresponding to primary works.

Round 2

Reviewer 1 Report

The authors incorporated most of the suggestions. I recommend publishing the manuscript with moderate English improvements.